# Novel Semi-Analytical Solutions for the Transient Behaviors of Functionally Graded Material Plates in the Thermal Environment

**DOI:** 10.3390/ma12244084

**Published:** 2019-12-06

**Authors:** Zeng Cao, Xu Liang, Yu Deng, Xing Zha, Ronghua Zhu, Jianxing Leng

**Affiliations:** Ocean College, Zhejiang University, Dinghai, Zhoushan 316021, China; zeng_cao@zju.edu.cn (Z.C.); dengyu_oe@zju.edu.cn (Y.D.); xing_zha@zju.edu.cn (X.Z.); zhu.richard@zju.edu.cn (R.Z.); jxleng@zju.edu.cn (J.L.)

**Keywords:** semi-analytical algorithm, transient behaviors, functionally graded material plate, in-plane displacements, thermal environment

## Abstract

The primary objective of this article is to present a semi-analytical algorithm for the transient behaviors of Functionally Graded Materials plates (FGM plates) considering both the influence of in-plane displacements and the influence of temperature changes. Based on the classical plate theory considering the effect of in-plane displacements, the equilibrium equations of the motion system are derived by Hamilton’s principle. Here, we propose a novel, accurate, and efficient semi-analytical method that incorporates the Fourier series expansion, the Laplace transforms, and its numerical inversion and the Differential Quadrature Method (DQM) to simulate the transient behaviors. This paper validates the proposed method by comparisons with semi-analytical natural frequency results and those from the literature. Expressly, the results of dynamic response also agree well with those generated by the Navier’s method and Finite Element Method (FEM). A convergence study that utilizes the different numbers of sampling points shows that the process can converge quickly, and a few sampling points can achieve high accuracy. The effects of various boundary conditions at the ends, material graded index, and temperature change are further investigated. From the detailed parametric study, it is seen that the peak displacement increases as the edge degrees of freedom, the gradient index of the material, and temperature change increase.

## 1. Introduction

The definition of Functionally Graded Materials (FGM), which was considered in Japan in 1984 firstly by materials scientists, is to make a class of composite materials that varies the material properties from one surface to the other based on a specific gradient [1,2,3]. In recent years, applications of FGM have become popular in structural engineering, mechanical engineering, and marine engineering because of the high performance heat resistant properties of FGM [4,5,6,7]. As the demands of FGM increase, especially Functionally Graded Material plates (FGM plates), FGM is often exposed to a variety of thermal conditions resulting in non-uniform and uniform thermal loads, which cause variation in the vibration characteristics of structures [8]. Thus, accurate and efficient tools will be required to predict their dynamic response for vibration control and structural design.

Given the efficient and safe use of FGM, many investigations considering multiple loading conditions and environments have been published by numerous research workers using different measures. Some of the papers have been carried out to understand the vibration characteristics of Functionally Graded FG plates such as the free vibration and stress distribution of a structure made of FGM [4,9,10,11,12,13]. However, these works ignored the effects of temperature, which is of great significance to FGM plates. Some research works related to FGM plates subjected to thermal loads are introduced in the following.

Malekzadeh [14,15] analyzed the natural frequencies of FGM plates considering the effect of temperature change employing the differential quadrature method and discussed the effects of different boundary conditions, temperature change, and the volume fraction index. Shenas [16] computed the free vibration of FG quadrilateral microplates subjected to thermal loads by the Chebyshev–Ritz method and concluded that the free frequencies increased monotonically with the temperature drops. Shi [17] chose 3D elasticity theory to compute the vibration characteristics of FGM annular plates employing the Chebyshev–Ritz algorithm and analyzed the influences of various mixed boundary conditions, the material graded index, and temperature rise on the eigenfrequencies. Pandey [18] applied the finite element formulation to analyze the natural frequencies of FGM sandwich plates considering the effect of temperature change and presented the influences of various boundary conditions and different FGM cores. Li [19] chose the Ritz method to propose three-dimensional results for the vibration characteristics of FGM sandwich plates in a thermal environment and suggested that the material properties could affect thin FG plates more significantly than thick FG plates. Wang [20] carried out nonlinear vibration for FGM sandwich plates in the thermal environment using a two step perturbation technique and presented that the variation in temperature had a significant influence on the fundamental frequency. Shen and Zhang [21] also analyzed the non-linear vibration of FGM composite laminated plates under different thermal conditions and found that fiber reinforcement resulted in a significant change in the vibration characteristic. Khalili [22] proposed a natural frequencies analysis for FGM sandwich plates in various thermal environments and presented a technique to reduce the governing equations of the motion system. He also found that temperature change significantly influenced the natural frequencies. George [23] used a finite element approach to compute the vibration characteristics of the FGM carbon reinforced polymer composite plate under thermal load and observed that the natural frequencies under a thermal environment increased as the temperature dropped. All the works cited above analyzed the free vibration of FGM plates in detail, but few studies have been done on the dynamic or transient behaviors of FGM plates. It can be seen that the majority of research mentioned above employed the Ritz technique to analyze the vibration characteristics of the structure. However, some drawbacks when using this kind of method, such as the difficulty of finding the trail functions and improving the numerical accuracy in some cases, can be found.

Huang and Shen [24] proposed an analytical method to simulate the nonlinear vibration and transient behaviors of FGM plates considering the thermal effect and discussed the impact of the volume fraction index and temperature field on transient responses. The displacement of an FGM plate subjected to transient thermal loading was studied by Kim [25] using the Galerkin method. The author found that the dimensionless displacement decreased as the volumetric ratio of metal decreased. Praveen and Reddy [26] investigated the static and transient behaviors of FGM plates by the finite element method and analyzed the influences of material properties and the variation in temperature on the dynamic behaviors of FGM plates. Malekzadeh [27] used Newmark’s time integration technique and the finite element method to analyze the dynamic response of FGM plates under moving and thermal load. Numerical methods such as the Galerkin method and finite element method have been proven to be efficient and precise techniques. However, numerical errors may occur because of spurious numerical oscillations [28,29]. From the literature concluded above, it also can be found that many types of research employed various plate theories, but few studies considered the effect of in-plane displacement.

While this summary shows that significant research has been carried out to analyze the free vibration and the dynamic response of FGM plates with various methods and plate theory, however, to the best of the author’s knowledge, little attention has been devoted to a semi-analytical algorithm for the dynamic behaviors of functionally graded materials plates considering both the influence of in-plane displacements and the effect of temperature changes. For the transient behavior of structures, there is an urgent need for a more efficient and accurate method. Given the circumstances, we propose a novel and accurate semi-analytical methodology that incorporates the Fourier series expansion, the Laplace transform, and its numerical inversion and the Differential Quadrature Method (DQM) to investigate the dynamic response of FGM plates. 

This paper is organized as follows. Section 2 derives the partial differential equations of the motion system using Hamilton’s principle. Section 3 demonstrates briefly the method we used in this work. A detailed numerical study including the validation and convergence study and the influences of different boundary conditions, the material graded index, and temperature change is presented in Section 4, followed by conclusions in the last section.

## 2. Theoretical Formulation

### 2.1. Material Properties

As shown in Figure 1, here, an FGM rectangular plate with length *a*, width *b*, and thickness *h* is considered. The FGM plates were mainly made of a mixture of metal and ceramic, which are used to manufacture the fuselage of the shuttle, the interior chamber walls and turbine engines such as the PSZ/IN 100 functionally graded material. The variation of the properties was smooth and continuous through the thickness by a simple power law distribution with Voigt’s rule of mixtures. Based on Voigt’s rule of combinations, the material properties *P*, such as Young’s modulus *E*, mass density *ρ*, Poisson’s ratio *v*, thermal conductivity *κ*, and thermal expansion coefficient *α*, are given by:(1)$$P=P_{c}+(P_{c}-P_{m}){\left(\frac{1}{2}+\frac{z}{h}\right)}^{n}$$
where *n* ≥ 0 is the power law index, which defines the distribution of the constituents in FGM and *P_c_* and *P_m_* denote the material properties of ceramic and metal, *z* represents *z*-axial coordinate, respectively.

The temperature distribution *T* in FGM plate can be computed by solving the one-dimensional steady-state heat transfer equation which assumes that the variation in temperature occurs in the z-axis direction [30]. The following equations can be used
(2)$$-\frac{d}{dz}[\kappa \frac{dT}{dz}]=0$$

The boundary condition can be expressed as:(3)$$T=T_{c}\;\mathrm{at}\;z=t_{1}\;(\mathrm{upper}\;\mathrm{surface})$$
(4)$$T=T_{m}\;\mathrm{at}\;z=t_{2}\;(\mathrm{lower}\;\mathrm{surface})$$

The variation in temperature can be obtained by solving Equations (2)–(4) using the technique of polynomial series:(5)$$T=\left(T_{c}-T_{m}\right)\eta \left(z\right)+T_{m}$$
in which:(6)$$\eta \left(z\right)=\frac{1}{C}\left[\begin{array}{l} \left(\frac{z}{h}+\frac{1}{2}\right)-\frac{k_{cm}}{k_{m}\left(n+1\right)}{\left(\frac{z}{h}+\frac{1}{2}\right)}^{n+1}+\frac{k_{cm}^{2}}{k_{m}^{2}\left(2n+1\right)}{\left(\frac{z}{h}+\frac{1}{2}\right)}^{2n+1}\\ -\frac{k_{cm}^{3}}{k_{m}^{3}\left(3n+1\right)}{\left(\frac{z}{h}+\frac{1}{2}\right)}^{3n+1}+\frac{k_{cm}^{4}}{k_{m}^{4}\left(4n+1\right)}{\left(\frac{z}{h}+\frac{1}{2}\right)}^{4n+1}-\\ \frac{k_{cm}^{5}}{k_{m}^{5}\left(5n+1\right)}{\left(\frac{z}{h}+\frac{1}{2}\right)}^{5n+1} \end{array}\right]$$
(7)$$\begin{array}{l} k_{cm}=k_{c}-k_{m},\\ C=1-\frac{k_{cm}}{\left(n+1\right)k_{m}}+\frac{k_{cm}^{2}}{\left(2n+1\right)k_{m}^{2}}-\frac{k_{cm}^{3}}{\left(3n+1\right)k_{m}^{3}}+\frac{k_{cm}^{4}}{\left(4n+1\right)k_{m}^{4}}-\frac{k_{cm}^{5}}{\left(5n+1\right)k_{m}^{5}} \end{array}$$
where *k_c_* and *k_m_* are the thermal conductivity of ceramic and thermal conductivity of metal.

In the high temperature environment, the properties of the constituent materials may change significantly. Therefore, there is a need for the thermal behavior of the FGM properties, and the following function [31] can be used as:(8)$$P_{i}=P_{0}\left(P_{-1}T^{-1}+1+P_{1}T+P_{2}T^{2}+P_{3}T^{3}\right)\left(i=c,m\right)$$
where *P_0_*, *P_−1_*, *P_1_*, *P_2_*, and *P_3_* are the given coefficients of temperature *T* (in K).

### 2.2. Mathematical Model

The mathematical model is proposed considering the influence of in-plane displacements:(9)$$\left\{\begin{array}{l} u(x,y,z,t)\\ v(x,y,z,t)\\ w(x,y,z,t) \end{array}\right\}=\left\{\begin{array}{l} u_{0}(x,y,t)-zw_{0,x}(x,y,t)\\ v_{0}(x,y,t)-zw_{0,y}(x,y,t)\\ w_{0}(x,y,t) \end{array}\right\}$$
where *t* is the time; *u_0_* and *v_0_* denote the mid-plane displacements of different directions (x-axis and y-axis); *w_0_* denotes the transverse displacements (z-axis).

The strain–displacement relations can be expressed as:(10)$$\varepsilon =\left\{\begin{array}{c} \varepsilon_{xx}\\ \varepsilon_{yy}\\ \varepsilon_{xy} \end{array}\right\}=\varepsilon_{0}+z\kappa ==\left\{\begin{array}{c} \frac{\partial u_{0}}{\partial x}\\ \frac{\partial v_{0}}{\partial y}\\ \frac{\partial u_{0}}{\partial y}+\frac{\partial v_{0}}{\partial x} \end{array}\right\}+z\left\{\begin{array}{c} -\frac{\partial^{2}w_{0}}{\partial x^{2}}\\ -\frac{\partial^{2}w_{0}}{\partial y^{2}}\\ -2\frac{\partial^{2}w_{0}}{\partial x\partial y} \end{array}\right\}$$
where ***ε***_0_ is the vector of strains on the middle surface and ***κ*** is the vector of curvature changes.

Based on Hooke’s law, the stress–strain relations can be expressed as:(11)$$\left\{\begin{array}{c} \sigma_{xx}\\ \sigma_{yy}\\ \sigma_{xy} \end{array}\right\}=\left[\begin{array}{l} \sigma_{xx}^{0}\\ \sigma_{yy}^{0}\\ \sigma_{xy}^{0} \end{array}\right]+\left[\begin{array}{l} \sigma_{xx}^{T}\\ \sigma_{yy}^{T}\\ \sigma_{xy}^{T} \end{array}\right]=\left[\begin{array}{ccc} Q_{11} & Q_{12} & 0\\ Q_{21} & Q_{22} & 0\\ 0 & 0 & Q_{66} \end{array}\right]\left[\left\{\begin{array}{c} \varepsilon_{xx}\\ \varepsilon_{yy}\\ \varepsilon_{xy} \end{array}\right\}-\Delta T\left\{\begin{array}{c} \alpha \\ \alpha \\ 0 \end{array}\right\}\right]$$
where Δ*T* and *α* are temperature changes and the thermal expansion coefficient, respectively; and the reduced stiffness *Q_ij_* (*i*, *j* = 1, 2, and 6) can be stated as:(12)$$\begin{array}{l} Q_{11}=Q_{22}=\frac{E\left(z\right)}{1-\nu^{2}\left(z\right)},\;\\ Q_{12}=Q_{21}=\frac{\nu \left(z\right)E\left(z\right)}{1-\nu^{2}\left(z\right)},\;\\ Q_{66}=\frac{E\left(z\right)}{2\left[1+\nu \left(z\right)\right]}. \end{array}$$
where *E* is the material elastic modulus; and *ν* is the material Poisson’s ratio.

Based on Equations (10)–(12), the governing equations of the motion system can be obtained by using Hamilton’s principle, which is expressed as:(13)$$\int_{0}^{t}\left(\delta T-\delta U+\delta W_{ek}\right)dt=0$$
where *T*, *U*, and *W_ek_* are the kinetic energy, the strain energy, and the work done by the external work, respectively.

The kinetic energy of the FGM plate can be stated as:(14)$$T=\frac{1}{2}{∭}_{V}\rho \left({\dot{u}}^{2}+{\dot{v}}^{2}+{\dot{w}}^{2}\right)dV$$
where the over-dot denotes the first derivative with respect of time, *ρ* is the density of the plate, *V* is the volume of the plate.

The strain energy of the FGM plate can be written as:(15)$$U=\frac{1}{2}{∭}_{V}(\sigma_{xx}\varepsilon_{xx}+\sigma_{yy}\varepsilon_{yy}+\sigma_{xy}\varepsilon_{xy})dV$$

The work done by external work can be expressed as:(16)$$W_{ek}=W_{ek}^{T}+W_{ek}^{F}$$
where *W^T^_ek_* and *W^F^_ek_* are external works done by temperature variation and external forces, which can be defined as:(17)$$W_{ek}^{T}=-\frac{1}{2}{∭}_{V}N_{xx}^{T}{\left(\frac{\partial w}{\partial x}\right)}^{2}+N_{yy}^{T}{\left(\frac{\partial w}{\partial y}\right)}^{2}dV$$
(18)$$W_{ek}^{F}={∭}_{V}f(x,y,t)wdV$$
where *f* (*x*, *y*, *t*) is the external force, and *N^T^* can be described by:(19)$$\{N_{xx}^{T},N_{yy}^{T}\}=\int_{-h/2}^{h/2}Q_{11}\alpha \Delta Tdz$$
where Δ*T* is the temperature change in the *z*-direction, which can be derived by Equations (5)–(7).

The governing equations of the motion system for the FGM rectangular plates can be derived by substituting Equations (14)–(16) into Equation (13); the following equations regarding the forces *N_ij_* and moment resultants *M_ij_* can be stated as:(20)$$\begin{array}{l} \frac{\partial N_{xx}}{\partial x}+\frac{\partial N_{xy}}{\partial y}=I_{0}\frac{\partial^{2}u_{0}}{\partial t^{2}}-I_{1}\frac{\partial^{3}w_{0}}{\partial t^{2}\partial x},\\ \frac{\partial N_{xy}}{\partial x}+\frac{\partial N_{yy}}{\partial y}=I_{0}\frac{\partial^{2}v_{0}}{\partial t^{2}}-I_{1}\frac{\partial^{3}w_{0}}{\partial t^{2}\partial y},\\ \frac{\partial^{2}M_{xx}}{\partial x^{2}}+2\frac{\partial^{2}M_{xy}}{\partial x\partial y}+\frac{\partial^{2}M_{yy}}{\partial y^{2}}=I_{0}\frac{\partial^{2}w_{0}}{\partial t^{2}}+I_{1}(\frac{\partial^{3}u_{0}}{\partial t^{2}\partial x}+\frac{\partial^{3}v_{0}}{\partial t^{2}\partial y})-I_{2}(\frac{\partial^{4}w_{0}}{\partial t^{2}\partial x^{2}}+\frac{\partial^{4}w_{0}}{\partial t^{2}\partial y^{2}}) \end{array}$$
where:(21)$$\begin{array}{l} \left\{N_{xx},N_{yy},N_{xy}\right\}=\int_{-h/2}^{h/2}\left\{\sigma_{xx},\sigma_{yy},\sigma_{xy}\right\}dz,\\ \left\{M_{xx},M_{yy},M_{xy}\right\}=\int_{-h/2}^{h/2}\left\{\sigma_{xx},\sigma_{yy},\sigma_{xy}\right\}zdz,\\ \{I_{0},I_{1},I_{2}\}=\int_{-h/2}^{h/2}\rho (z)(1,z,z^{2})dz. \end{array}$$

By substituting Equation (21) into Equation (20), the governing equations of the motion system can be obtained as:(22)$$A_{11}\frac{\partial^{2}u_{0}}{\partial x^{2}}+A_{12}\frac{\partial^{2}v_{0}}{\partial x\partial y}+A_{33}(\frac{\partial^{2}u_{0}}{\partial y^{2}}+\frac{\partial^{2}v_{0}}{\partial x\partial y})-B_{11}\frac{\partial^{3}w_{0}}{\partial x^{3}}\;-B_{12}\frac{\partial^{3}w_{0}}{\partial x\partial y^{2}}-2B_{33}\frac{\partial^{3}w_{0}}{\partial x\partial y^{2}}=I_{0}\frac{\partial^{2}u_{0}}{\partial t^{2}}-I_{1}\frac{\partial^{3}w_{0}}{\partial t^{2}\partial x},\;A_{11}\frac{\partial^{2}v_{0}}{\partial y^{2}}+A_{12}\frac{\partial^{2}u_{0}}{\partial x\partial y}+A_{33}(\frac{\partial^{2}v_{0}}{\partial x^{2}}+\frac{\partial^{2}u_{0}}{\partial x\partial y})-B_{11}\frac{\partial^{3}w_{0}}{\partial y^{3}}\;-B_{12}\frac{\partial^{3}w_{0}}{\partial x^{2}\partial y}-2B_{33}\frac{\partial^{3}w_{0}}{\partial x^{2}\partial y}=I_{0}\frac{\partial^{2}v_{0}}{\partial t^{2}}-I_{1}\frac{\partial^{3}w_{0}}{\partial t^{2}\partial y},\;B_{11}(\frac{\partial^{3}v_{0}}{\partial y^{3}}+\frac{\partial^{3}u_{0}}{\partial x^{3}})+B_{12}(\frac{\partial^{3}u_{0}}{\partial x\partial y^{2}}+\frac{\partial^{3}v_{0}}{\partial x^{2}\partial y})+B_{33}(2\frac{\partial^{3}u_{0}}{\partial x\partial y^{2}}+2\frac{\partial^{3}v_{0}}{\partial x^{2}\partial y})\;-D_{11}(\frac{\partial^{4}w_{0}}{\partial x^{4}}+\frac{\partial^{4}w_{0}}{\partial y^{4}})-2D_{12}\frac{\partial^{4}w_{0}}{\partial x^{2}\partial y^{2}}-4D_{33}\frac{\partial^{4}w_{0}}{\partial x^{2}\partial y^{2}}-N_{xx}^{T}\frac{\partial^{2}w_{0}}{\partial x^{2}}-N_{yy}^{T}\frac{\partial^{2}w_{0}}{\partial y^{2}}\;=I_{0}\frac{\partial^{2}w_{0}}{\partial t^{2}}+I_{1}(\frac{\partial^{3}u_{0}}{\partial t^{2}\partial x}+\frac{\partial^{3}v_{0}}{\partial t^{2}\partial y})-I_{2}(\frac{\partial^{4}w_{0}}{\partial t^{2}\partial x^{2}}+\frac{\partial^{4}w_{0}}{\partial t^{2}\partial y^{2}})-f(x,y,t)$$
where:(23)$$\left\{A_{ij},B_{ij},D_{ij}\right\}=\int_{-h/2}^{h/2}\left\{1,z,z^{2}\right\}Q_{ij}dz,\;(i,j=1,2,\ldots 6).$$

As shown in Figure 2, consider various boundary conditions of FGM rectangular plates here: S-S-S-S; S-S-S-C; S-S-C-C; S-S-S-F. S denotes Simply supported; C denotes Clamped; F denotes Free.

For a simply supported end, it can be stated as:(24)$$v=w=N_{yy}=M_{yy}=0$$

For a clamped end, it can be stated as:(25)$$u=v=w=\frac{\partial w}{\partial x}=0$$

For a free end, it can be expressed as:(26)$$M_{xx}=M_{xy}=N_{xx}=V_{xx}$$
where: $$V_{xx}=\frac{\partial M_{xx}}{\partial x}+2\frac{\partial M_{xy}}{\partial y}$$

## 3. Solution Procedure

### 3.1. Differential Quadrature Method and Laplace Transform

#### 3.1.1. Differential Quadrature Method

The differential method (DQM) has been proven to be able to solve various structural problems efficiently and accurately [32]. The Differential Quadrature Method (DQM) can approximate the partial derivative of a function by a weighted linear sum of the function values as below [33]:(27)$$\frac{\partial^{n}f(x_{i})}{\partial x^{n}}=\sum_{i=1}^{N}W_{ij}^{\left(n\right)}f(x_{j})(i,j=1,2,\ldots ,N)$$
where *f*(*x*) denotes a continuous function, *N* is the number of discrete sampling points, $W_{(n)\;ij}$ denotes the weighting coefficients, *x_i,j,k_* are the given points, and *N* denotes a large integer. The weighting coefficients $W_{(n)\;ij}$ can be derived by:(28)$$W_{ij}^{\left(1\right)}=\frac{\prod_{k=1,k\neq i}^{N}x_{i}-x_{k}}{(x_{j}-x_{i})\prod_{k=1,k\neq j}^{N}x_{j}-x_{k}},i\neq j\;(i,j=1,2,3,\ldots ,N);\;W_{ij}^{\left(1\right)}=\frac{1}{x_{j}-x_{i}}\prod_{\begin{array}{l} k=1\\ k\neq i \end{array}}^{N}\frac{1}{x_{i}-x_{k}},\;i=j\;(i=1,2,3,\ldots ,N);\;W_{mn}^{\left(k\right)}=k(W_{ij}^{\left(k-1\right)}W_{ij}^{\left(1\right)}-\frac{W_{ij}^{\left(k-1\right)}}{x_{i}-x_{j}}),\;i\neq j\;(i,j=1,2,3,\ldots ,N;k=1,2,\ldots ,N-1);\;\;W_{ij}^{\left(k\right)}=-\sum_{i=1\;i\neq j}^{N}W_{ij}^{\left(k\right)},\;i=j,\;(i=1,2,3,\ldots ,N,\;k=1,2,\ldots ,N-1)$$

#### 3.1.2. Laplace Numerical Inversion

Durbin proposed the numerical inversion of the Laplace transform. This kind of method, which has been widely used in various engineering fields, can obtain almost the same solutions as that given by analytical inversion in a shorter time [34,35]. In that case, it can be defined by:(29)f(t→)≅2eλt→T{f˜(λ)2+∑m=1MRe[f˜(λ+mπi/B)]cos(mπt→/B)}
where *f*(*t*) is a function, *t* is the time, $\tilde{f}\left(\lambda \right)$ is the transformed function; *λ* = 5/*B*, *T* = 5 × *B_d_*, *B_d_* is the observing-time period, and *M* is a large integer.

### 3.2. Dynamic Response Calculation

This section derives the semi-analytical results using the Laplace transform and DQM for the dynamic behaviors with various boundary conditions in the thermal environment.

Firstly, the displacements on the middle face and external pressure can be expanded by the trigonometric Fourier series of the y-direction, as shown below:(30)$$\begin{array}{ll} u_{0}(x,y,t)=\sum_{m=1}^{\infty }{\bar{u}}_{0}(x,t)\sin\frac{m\pi y}{b},\; & v_{0}(x,y,t)=\sum_{m=1}^{\infty }{\bar{v}}_{0}(x,t)\cos\frac{m\pi y}{b},\\ w_{0}(x,y,t)=\sum_{m=1}^{\infty }{\bar{w}}_{0}(x,t)\sin\frac{m\pi y}{b},\; & f(x,y,t)=\sum_{m=1}^{\infty }\bar{f}(x,t)\sin\frac{m\pi y}{b}, \end{array}$$
where $\bar{f}\left(x,t\right)$ can be stated as:(31)$$\bar{f}(x,t)=\frac{2f(x,y,t)(1-\cos m\pi )}{m\pi }$$

By substituting Equations (30) and (31) into Equation (22) and utilizing the Laplace transform, the differential equations can be yielded:(32)$$(\frac{A_{33}m^{2}\pi^{2}}{b^{2}}+I_{0}\xi^{2}){\tilde{\bar{u}}}_{0}(x)+(\frac{A_{12}m\pi }{b}+\frac{A_{33}m\pi }{b})\frac{\partial {\tilde{\bar{v}}}_{0}(x)}{\partial x}-A_{11}\frac{\partial^{2}{\tilde{\bar{u}}}_{0}(x)}{\partial x^{2}}\;-(\frac{B_{12}m^{2}\pi^{2}}{b^{2}}+\frac{2B_{33}m^{2}\pi^{2}}{b^{2}}+I_{1}\xi^{2})\frac{\partial {\tilde{\bar{w}}}_{0}(x)}{\partial x}+B_{11}\frac{\partial^{3}{\tilde{\bar{w}}}_{0}(x)}{\partial x^{3}}=0,\;(\frac{A_{11}m^{2}\pi^{2}}{b^{2}}+I_{0}\xi^{2}){\tilde{\bar{v}}}_{0}(x)-(\frac{B_{11}m^{3}\pi^{3}}{b^{3}}+\frac{I_{1}\xi^{2}m\pi }{b}){\tilde{\bar{w}}}_{0}(x)-A_{33}\frac{\partial^{2}{\tilde{\bar{v}}}_{0}(x)}{\partial x^{2}}\;-(\frac{A_{12}m\pi }{b}+\frac{A_{33}m\pi }{b})\frac{\partial {\tilde{\bar{u}}}_{0}(x)}{\partial x}+(\frac{B_{12}m\pi }{b}+\frac{2B_{33}m\pi }{b})\frac{\partial^{2}{\tilde{\bar{w}}}_{0}(x)}{\partial x^{2}}=0,\;-(\frac{B_{11}m^{3}\pi^{3}}{b^{3}}+\frac{I_{1}\xi^{2}m\pi }{b}){\tilde{\bar{v}}}_{0}(x)+(-\frac{N_{yy}^{T}m^{2}\pi^{2}}{b^{2}}+\frac{D_{11}m^{4}\pi^{4}}{b^{4}}+I_{0}\xi^{2}+\frac{I_{2}\xi^{2}m^{2}\pi^{2}}{b^{2}}){\tilde{\bar{w}}}_{0}(x)\;{\tilde{\bar{w}}}_{0}(x)+(\frac{B_{12}m^{2}\pi^{2}}{b^{2}}+\frac{2B_{33}m^{2}\pi^{2}}{b^{2}})\frac{\partial {\tilde{\bar{u}}}_{0}(x)}{\partial x}+I_{1}\xi^{2}\frac{\partial {\tilde{\bar{u}}}_{0}(x)}{\partial x}\;+(\frac{B_{12}m\pi }{b}+\frac{2B_{33}m\pi }{b})\frac{\partial^{2}{\tilde{\bar{v}}}_{0}(x)}{\partial x^{2}}+(N_{xx}^{T}-\frac{2D_{12}m^{2}\pi^{2}}{b^{2}}-\frac{4D_{33}m^{2}\pi^{2}}{b^{2}}-I_{2}\xi^{2})\frac{\partial^{2}{\tilde{\bar{w}}}_{0}(x)}{\partial x^{2}}\;-B_{11}\frac{\partial^{3}{\tilde{\bar{u}}}_{0}(x)}{\partial x^{3}}+D_{11}\frac{\partial^{4}{\tilde{\bar{w}}}_{0}(x)}{\partial x^{4}}-\tilde{\bar{f}}(x)=0.$$

Employing the DQM on Equation (32) gives:(33)$$(\frac{A_{33}m^{2}\pi^{2}}{b^{2}}+I_{0}\xi^{2}){\tilde{\bar{u}}}_{0i}+(\frac{A_{12}m\pi }{b}+\frac{A_{33}m\pi }{b})\sum_{j=1}^{N}W_{ij}^{\left(1\right)}{\tilde{\bar{v}}}_{0j}-A_{11}\sum_{j=1}^{N}W_{ij}^{\left(2\right)}{\tilde{\bar{u}}}_{0j}\;-(\frac{B_{12}m^{2}\pi^{2}}{b^{2}}+\frac{2B_{33}m^{2}\pi^{2}}{b^{2}}+I_{1}\xi^{2})\sum_{j=1}^{N}W_{ij}^{\left(1\right)}{\tilde{\bar{w}}}_{0j}+B_{11}\sum_{j=1}^{N}W_{ij}^{\left(3\right)}{\tilde{\bar{w}}}_{0j}=0,\;(\frac{A_{11}m^{2}\pi^{2}}{b^{2}}+I_{0}\xi^{2}){\tilde{\bar{v}}}_{0i}-(\frac{B_{11}m^{3}\pi^{3}}{b^{3}}+\frac{I_{1}\xi^{2}m\pi }{b}){\tilde{\bar{w}}}_{0i}-A_{33}\sum_{j=1}^{N}W_{ij}^{\left(2\right)}{\tilde{\bar{v}}}_{0j}\;-(\frac{A_{12}m\pi }{b}+\frac{A_{33}m\pi }{b})\sum_{j=1}^{N}W_{ij}^{\left(1\right)}{\tilde{\bar{u}}}_{0j}+(\frac{B_{12}m\pi }{b}+\frac{2B_{33}m\pi }{b})\sum_{j=1}^{N}W_{ij}^{\left(2\right)}{\tilde{\bar{w}}}_{0j}=0,\;-(\frac{B_{11}m^{3}\pi^{3}}{b^{3}}+\frac{I_{1}\xi^{2}m\pi }{b}){\tilde{\bar{v}}}_{0j}+(-\frac{N_{yy}^{T}m^{2}\pi^{2}}{b^{2}}+\frac{D_{11}m^{4}\pi^{4}}{b^{4}}+I_{0}\xi^{2}+\frac{I_{2}\xi^{2}m^{2}\pi^{2}}{b^{2}}){\tilde{\bar{w}}}_{0j}\;+(\frac{B_{12}m^{2}\pi^{2}}{b^{2}}+\frac{2B_{33}m^{2}\pi^{2}}{b^{2}}+I_{1}\xi^{2})\sum_{j=1}^{N}W_{ij}^{\left(1\right)}{\tilde{\bar{u}}}_{0j}+(\frac{B_{12}m\pi }{b}+\frac{2B_{33}m\pi }{b})\sum_{j=1}^{N}W_{ij}^{\left(2\right)}{\tilde{\bar{v}}}_{0j}\;+(N_{xx}^{T}-\frac{2D_{12}m^{2}\pi^{2}}{b^{2}}-\frac{4D_{33}m^{2}\pi^{2}}{b^{2}}-I_{2}\xi^{2})\sum_{j=1}^{N}W_{ij}^{\left(2\right)}{\tilde{\bar{w}}}_{0j}-B_{11}\sum_{j=1}^{N}W_{ij}^{\left(3\right)}{\tilde{\bar{u}}}_{0j}\;+D_{11}\sum_{j=1}^{N}W_{ij}^{\left(4\right)}{\tilde{\bar{w}}}_{0j}-\tilde{\bar{f}}(x_{j})=0.$$

For simplicity, Equation (33) can be written in matrix form as:(34)S•W=F
where:(35)$$\begin{array}{c} U=\left\{\begin{array}{c} u_{0}\\ v_{0}\\ w_{0} \end{array}\right\},F=\left\{\begin{array}{c} 0\\ 0\\ f \end{array}\right\},S=\left\{\begin{array}{ccc} S_{11} & S_{12} & S_{13}\\ S_{21} & S_{22} & S_{23}\\ S_{31} & S_{32} & S_{33} \end{array}\right\},\\ u_{0}^{T}=\left\{{\tilde{\bar{u}}}_{01},\ldots {\tilde{\bar{u}}}_{0j},\ldots {\tilde{\bar{u}}}_{0N}\right\},v_{0}^{T}=\left\{{\tilde{\bar{v}}}_{01},\ldots {\tilde{\bar{v}}}_{0j},\ldots {\tilde{\bar{v}}}_{0N}\right\},\\ w_{0}^{T}=\left\{{\tilde{\bar{w}}}_{01},\ldots {\tilde{\bar{w}}}_{0j},\ldots {\tilde{\bar{w}}}_{0N}\right\},f=\left\{{\tilde{\bar{f}}}_{1},\ldots {\tilde{\bar{f}}}_{j},\ldots {\tilde{\bar{f}}}_{N}\right\},\\ I_{N}={\left[\begin{array}{cccc} 1 & 0 & \cdots & 0\\ 0 & 1 & \cdots & 0\\ \vdots & \vdots & \ddots & \vdots \\ 0 & 0 & \cdots & 1 \end{array}\right]}_{N\times N},W_{ij}^{\left(m\right)}=\left[\begin{array}{cccc} W_{11}^{\left(m\right)} & W_{12}^{\left(m\right)} & \cdots & W_{1N}^{\left(m\right)}\\ W_{21}^{\left(m\right)} & W_{22}^{\left(m\right)} & \cdots & W_{2N}^{\left(m\right)}\\ \vdots & \vdots & \ddots & \vdots \\ W_{N1}^{\left(m\right)} & W_{N2}^{\left(m\right)} & \cdots & W_{NN}^{\left(m\right)} \end{array}\right] \end{array}$$

(36)$$\begin{array}{ll} S_{11}= & (\frac{A_{33}m^{2}\pi^{2}}{b^{2}}+I_{0}\xi^{2})I_{N}-A_{11}W_{ij}^{\left(2\right)},\\ S_{12}= & (\frac{A_{12}m\pi }{b}+\frac{A_{33}m\pi }{b})W_{ij}^{\left(1\right)},\\ S_{13}= & -(\frac{B_{12}m^{2}\pi^{2}}{b^{2}}+\frac{2B_{33}m^{2}\pi^{2}}{b^{2}}+I_{1}\xi^{2})W_{ij}^{\left(1\right)},\\ S_{21}= & -(\frac{A_{12}m\pi }{b}+\frac{A_{33}m\pi }{b})W_{ij}^{\left(1\right)},\\ S_{22}= & (\frac{A_{11}m^{2}\pi^{2}}{b^{2}}+I_{0}\xi^{2})I_{N}-A_{33}W_{ij}^{\left(2\right)},\\ S_{23}= & -(\frac{B_{11}m^{3}\pi^{3}}{b^{3}}+\frac{I_{1}\xi^{2}m\pi }{b})I_{N}+(\frac{B_{12}m\pi }{b}+\frac{2B_{33}m\pi }{b})W_{ij}^{\left(2\right)},\\ S_{31}= & (\frac{B_{12}m^{2}\pi^{2}}{b^{2}}+\frac{2B_{33}m^{2}\pi^{2}}{b^{2}}+I_{1}\xi^{2})W_{ij}^{\left(1\right)}-B_{11}W_{ij}^{\left(3\right)},\\ S_{32}= & (\frac{B_{12}m\pi }{b}+\frac{2B_{33}m\pi }{b})W_{ij}^{\left(2\right)}-(\frac{B_{11}m^{3}\pi^{3}}{b^{3}}+\frac{I_{1}\xi^{2}m\pi }{b})I_{N},\\ S_{33}= & (-\frac{N_{yy}^{T}m^{2}\pi^{2}}{b^{2}}+\frac{D_{11}m^{4}\pi^{4}}{b^{4}}+I_{0}\xi^{2}+\frac{I_{2}\xi^{2}m^{2}\pi^{2}}{b^{2}})I_{N}\\ & +(N_{xx}^{T}-\frac{2D_{12}m^{2}\pi^{2}}{b^{2}}-\frac{4D_{33}m^{2}\pi^{2}}{b^{2}}-I_{2}\xi^{2})W_{ij}^{\left(2\right)}+D_{11}W_{ij}^{\left(4\right)}. \end{array}$$

Several efficient and accurate methods can be used to consider the boundary conditions. Here, we consider FGM rectangular plates with clamped ends by equation substitution. Similarly, the corresponding conditions can be expressed as:(37)$$\begin{array}{l} {\tilde{\bar{u}}}_{01}={\tilde{\bar{v}}}_{01}={\tilde{\bar{w}}}_{01}=\sum_{j=1}^{N}W_{1j}^{\left(1\right)}{\tilde{\bar{w}}}_{01}=0,\;x=0,\\ {\tilde{\bar{u}}}_{0N}={\tilde{\bar{v}}}_{0N}={\tilde{\bar{w}}}_{0N}=\sum_{j=1}^{N}W_{Nj}^{\left(1\right)}{\tilde{\bar{w}}}_{0N}=0,\;x=a. \end{array}$$

After substituting Equation (37) into (34), the matrix **S** is rewritten as:(38)$$\begin{array}{l} S_{ii}=\left[\begin{array}{ccccc} 1 & 0 & \cdots & 0 & 0\\ S_{ii}^{21} & S_{ii}^{22} & \cdots & S_{ii}^{2,N-1} & S_{ii}^{2N}\\ \vdots & \vdots & \ddots & \vdots & \vdots \\ S_{ii}^{N-1,2} & S_{ii}^{N-1,2} & \cdots & S_{ii}^{N-1,N-1} & S_{ii}^{N-1,N}\\ 0 & 0 & \cdots & 0 & 1 \end{array}\right],\left(i=1,2\right)\\ S_{ij}=\left[\begin{array}{ccccc} 0 & 0 & \cdots & 0 & 0\\ S_{ij}^{21} & S_{ij}^{22} & \cdots & S_{ij}^{2,N-1} & S_{ij}^{2N}\\ \vdots & \vdots & \ddots & \vdots & \vdots \\ S_{ij}^{N-1,2} & S_{ij}^{N-1,2} & \cdots & S_{ij}^{N-1,N-1} & S_{ij}^{N-1,N}\\ 0 & 0 & \cdots & 0 & 0 \end{array}\right],\left(i,j=1,2,3;i\neq j\right)\\ S_{33}=\left[\begin{array}{ccccccc} 1 & 0 & 0 & \cdots & 0 & 0 & 0\\ W_{11}^{\left(1\right)} & W_{12}^{\left(1\right)} & W_{13}^{\left(1\right)} & \cdots & W_{1,N-2}^{\left(1\right)} & W_{1,N-1}^{\left(1\right)} & W_{1N}^{\left(1\right)}\\ S_{33}^{31} & S_{33}^{32} & S_{33}^{33} & \cdots & S_{33}^{3,N-2} & S_{33}^{3,N-1} & S_{33}^{3N}\\ \vdots & \vdots & \vdots & \ddots & \vdots & \vdots & \vdots \\ S_{33}^{N-3,1} & S_{33}^{N-3,2} & S_{33}^{N-3,3} & \cdots & S_{33}^{N-3,N-2} & S_{33}^{N-3,N-1} & S_{33}^{N-3,N}\\ W_{N1}^{\left(1\right)} & W_{N2}^{\left(1\right)} & W_{N3}^{\left(1\right)} & \cdots & W_{N,N-2}^{\left(1\right)} & W_{N,N-1}^{\left(1\right)} & W_{NN}^{\left(1\right)}\\ 0 & 0 & 0 & \cdots & 0 & 0 & 1 \end{array}\right] \end{array}$$

We can find the nontrivial solution in the Laplace transformed domain by solving Equation (34). Subsequently, the numerical inversion of the Laplace transform can be applied to obtain the solution in the time domain.

## 4. Numerical Results

This section presents the numerical results based on a proposed semi-analytical method for the transient response of the FGM rectangular plates in the thermal environment. A software package was designed to evaluate the model in this work using the software Wolfram Mathematica [36]. Firstly, a section carried out the convergence of the results. Subsequently, validation of the proposed semi-analytical method was carried out, which included the contrast of fundamental frequencies and a comparison of the dynamic response. Lastly, we assessed the influences of various boundary conditions, temperature change, and the material graded index on the dynamic response. Considering a particular FGM model, which is made of Si_3_N_4_ (ceramic) and SUS304 (metal), the material properties of the model are listed in Table 1 [37]. The distributed excitation was considered as f(x,y,t)=Q Sin(30t). The excitation of the model that is listed in Table 2 was supposed to apply to the upper surface (metal).

The choice of the sampling point system is crucial for computation [38]. In the present work, the sampling point system can be designed as:(39)$$x_{1}=0,\;x_{2}=0.0001a,\;x_{j}=\frac{a}{2}\left[1-\cos\frac{\pi (j-2)}{N-3}\right],\;x_{N-1}=0.9999a,\;x_{N}=a.$$

### 4.1. Convergence Studies

To demonstrate the accuracy of the methodology, a series of different numbers of sampling points, *N* = 5, 7, 9, and 11, was studied. As shown in Table 2, this section used FGM rectangular plates under the S-S-S-S boundary condition and for different geometries, thermal environments, and power law indexes. Figure 3 shows the time histories of the transverse displacement at one given position (*x* = *a*/2, *y* = *b*/2). The figures of the four cases demonstrated that the same time histories could be predicted when *N* was larger than nine, showing that the semi-analytical methodology converged rapidly with increasing the sampling points.

### 4.2. Validation of the Proposed Method

#### 4.2.1. Validation by Comparison with Natural Frequencies Results

This validation compared the natural frequencies results obtained by the proposed method with those results from the literature [39]. Table 3 presents a comparison of the results of fundamental frequencies for four different boundary conditions (S-S-S-S, S-C-S-C, S-S-S-C, S-S-S-F) with different power law indexes (*n* = 0, 0.5, 1, 2). It was evident that the results obtained by the semi-analytical method agreed well with the results from Baferani and Jomehzadeh [39], no matter what law of the variation of properties or boundary condition was utilized.

#### 4.2.2. Validation by Comparison with Analytical and FEM Solutions

For further validation, we validated the presented semi-analytical method by comparing it with solutions obtained by using Navier’s method and commercial software COMSOL (based on FEM). Consider FGM rectangular plates with the boundary condition S-S-S-S in this section. Four cases, as shown in Table 2, were used here. Figure 4 compares the results of the dynamic response of FGM rectangular plates at one given position (*x* = *a*/2, *y* = *b*/2) obtained by Navier’s method, FEM, and the presented method. As shown in this figure, the solutions derived by the three methods coincided with each other, and the present method was well validated in the four cases.

To demonstrate the computational efficiency of the semi-analytical method, the same computer system and the three methods (the proposed method, Navier’s method, and FEM) were employed to analyze the four cases, as shown in Table 2. The relevant Computation Times (CT) and computer configuration instructions are listed in Table 4. It can be seen that the semi-analytical method can use a shorter simulation time than the FEM to obtain the results. The time using Navier’s method was longer than the proposed method when the sampling numbers *N* = 7 and was close to the semi-analytical method when the sampling numbers *N* = 9. However, Navier’s method could obtain the exact results with only suitable boundary conditions, such as simply supported at both edges. Therefore, by comparing with two widely used and classical methods, the present method had a high simulation efficiency.

### 4.3. The Influence of the Boundary Conditions at the Ends

This section studies the dynamic response due to the impacts of the boundary conditions. The various parameters (geometry, the parameters of the thermal environment, material graded index, and load parameter) used to simulate were the same as those in Case 1 (Table 2). Four different boundary conditions at the ends were considered here: S-S-S-S, S-S-S-C, S-S-C-C, S-S-S-F. Figure 5 demonstrates the dynamic behaviors of FGM rectangular plates at one given position (*x* = *a*/2, *y* = *b*/2). It can be concluded that the peak displacements of the FGM plates constantly decreased when the total number of degrees of freedom decreased (where *D*_S-S-S-F_ > *D*_S-S-S-S_ > *D*_S-S-S-C_ > *D*_S-S-C-C_, where *D* denotes the edge of the FGM rectangular plates and the subscripts denote the various boundary conditions).

### 4.4. The Influence of Material Graded Index n

This section analyzes the dynamic behaviors due to the impacts of material graded index *n*. The geometry, the load parameter, and the parameters of the thermal environment used for numerical computations were the same as those in Case 1 (Table 2). The FGM rectangular plates under S-S-S-S, S-S-S-C, S-S-C-C, and S-S-S-F boundary conditions were considered for the simulation. A set of FGM material graded indexes *n* = 0, 1/2, 1, and 2 was employed. Figure 6 shows the time histories of the transverse displacement at one given position (*x* = *a*/2, *y* = *b*/2). As is illustrated in the line chart, the displacement of the FGM plates decreased as the material graded index *n* increased, no matter which boundary conditions were employed.

### 4.5. The Influence of Temperature Changes

This section examines the influences of temperature changes on the transient response of FGM plates. The geometry, load parameter, and the material graded index used to simulate here were the same as those in Case 1 (Table 2). Consider the boundary conditions here: S-S-S-S, S-S-S-C, S-S-C-C, S-S-S-F. The simulation was implemented in four different thermal states with a structure temperature change of 0 K, 200 K, 500 K, and 700 K, as shown in Figure 7. It was obvious that the deflection of the FGM plates increased as the temperature increased. According to Equation (19), the axial tensile or compressive forces could be generated because of the sudden temperature change, which led to the peak deflection increases [40].

## 5. Conclusions

The transient behavior of FGM plates subjected to thermal loading was studied using a novel semi-analytical method in this article. Hamilton’s principle was employed to derive the formulation, which considered the effect of in-plane displacements. The plate was supposed to vary continuously in the thickness direction, and Voigt’s rule of mixtures was employed to estimate the material properties. This article presents a novel, accurate, and efficient semi-analytical method that incorporated the Laplace transform and its numerical inversion, Fourier series expansion technology, and the differential quadrature method (DQM) to analyze the transient behaviors of FGM plates.

Firstly, this paper validated the proposed method by a comparison of fundamental frequencies with those in the literature on the subject. For further validation, this article also carried out comparisons of the present solutions with those derived by Navier’s method. It all showed that the proposed semi-analytical method was accurate and efficient. Secondly, the convergence study that was carried out in this paper showed that the process had a fast convergence rate and a few sampling points could achieve high accuracy. Finally, the influences of various boundary conditions, the gradient index of the material, and temperature change were investigated. From the detailed parametric study, the peak displacements increased with the edge degrees of freedom, the gradient index of the material, and temperature change increase. The results obtained here are useful in the dynamic structural field.

## Figures and Tables

**Figure 1 materials-12-04084-f001:**
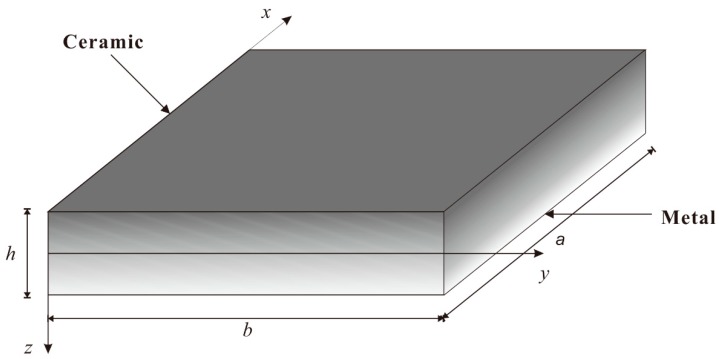
Geometry of the Functionally Graded Materials (FGM) rectangular plate. The length, width, and thickness of the FGM rectangular plate are *a*, *b*, and *h*, respectively. The variation of the properties of the FGM plate is assumed to be a power law distribution with Voigt’s rule of mixtures. The material is assumed to vary from ceramic to metal through the thickness. (x, y and z represent the transverse direction, longitudinal direction and vertical direction of rectangular coordinates.)

**Figure 2 materials-12-04084-f002:**
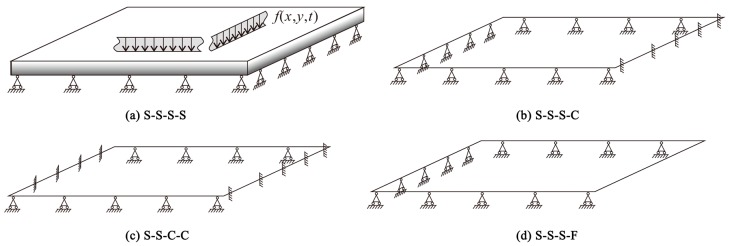
The schematic diagram of the boundary conditions. (**a**) S-S-S-S; (**b**) S-S-S-C; (**c**) S-S-C-C; (**d**) S-S-S-F. The arrows illustrate the distributed load on the metal surface of the FGM plates. The triangles represent the Simply (S) support boundary conditions. The vertical lines illustrate the Clamp (C) boundary conditions.

**Figure 3 materials-12-04084-f003:**
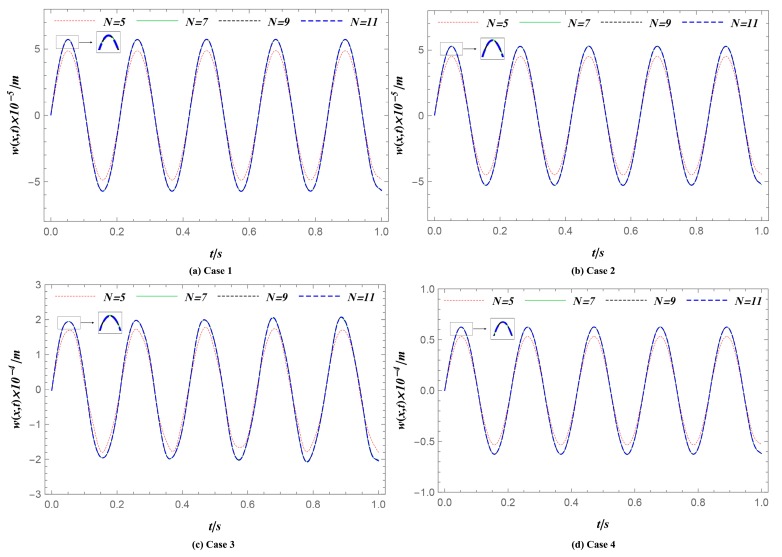
Deflection history *w* at one given position (*x* = *a*/2, *y* = *b*/2) for the convergence study. (**a**) Case 1: reference group. (**b**) Case 2: increasing power index *n*. (**c**) Case 3: decreasing thickness *h*. (**d**) Case 4: increasing temperature *T_c_*. The deflection histories *w* can converge when sampling points number *N* ≥ 9 in four different cases.

**Figure 4 materials-12-04084-f004:**
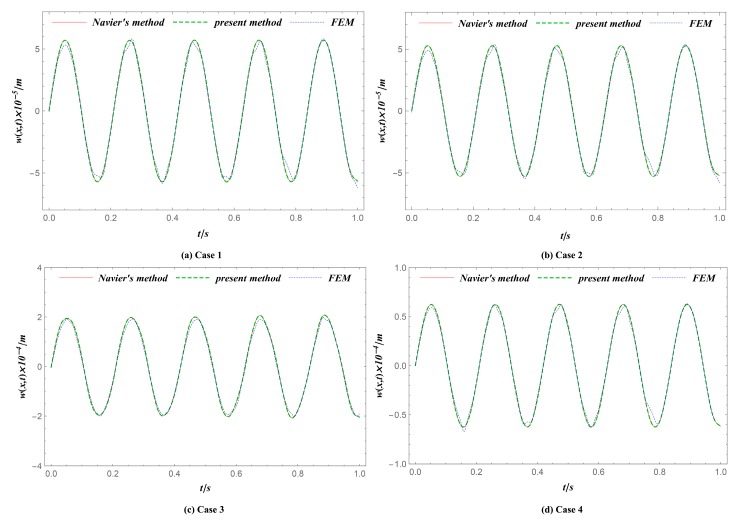
Deflection history *w* at one given position (*x* = *a*/2, *y* = *b*/2) obtained by the proposed approach, Navier’s algorithm, and FEM. (**a**) Case 1: reference group. (**b**) Case 2: increasing power index *n*. (**c**) Case 3: decreasing thickness *h*. (**d**) Case 4: increasing temperature *T_c_*. The solutions predicted by the three methods have a good agreement.

**Figure 5 materials-12-04084-f005:**
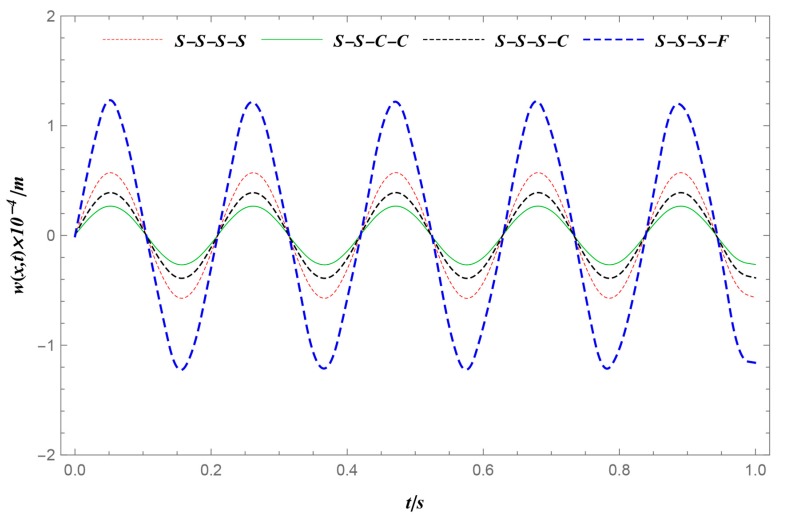
Deflection history *w* at one given position (*x* = *a*/2, *y* = *b*/2) under different boundary conditions. The peak deflection of the FGM plate decreases when the total number of degrees of freedom decreases.

**Figure 6 materials-12-04084-f006:**
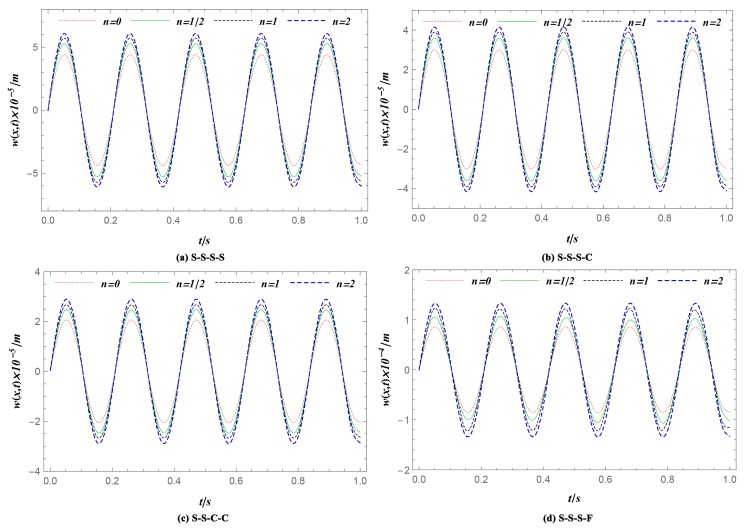
Deflection history *w* at one given position (*x* = *a*/2, *y* = *b*/2) with different material graded index *n*. (**a**) S-S-S-S; (**b**) S-S-S-C; (**c**) S-S-C-C; (**d**) S-S-S-F. The peak deflection of FGM plates decreases as the material graded index *n* increases, no matter which boundary conditions are employed.

**Figure 7 materials-12-04084-f007:**
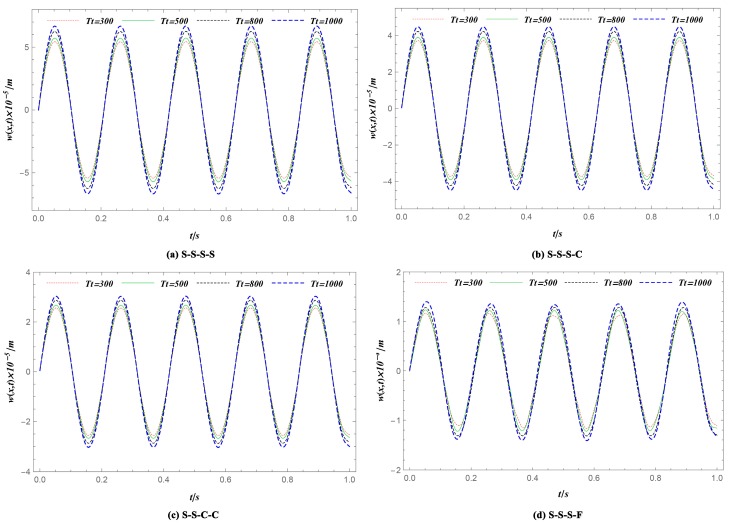
Deflection history *w* at one given position (*x* = *a*/2, *y* = *b*/2) with different temperature changes. (**a**) S-S-S-S; (**b**) S-S-S-C; (**c**) S-S-C-C; (**d**) S-S-S-F. The peak deflection of FGM plates increases as the temperature increases. The peak displacement increases by 6.5% when the temperature increases by 200 K.

**Table 1 materials-12-04084-t001:** Material properties of FGM components in the present work.

Material	Properties	P_0_	P_−1_	P_1_	P_2_	P_3_
Si_3_N_4_ (Ceramic)	*E* (Pa)	3.84 × 10^11^	0	−3.70 × 10^−4^	2.16 × 10^−7^	−8.95 × 10^−11^
	*ρ* (kg/m^3^)	2370	0	0	0	0
	*α* (K^−1^)	5.87 × 10^−6^	0	9.10 × 10^−4^	0	0
	*K* (W/mK)	13.723	0	0	0	0
	*ν*	0.24	0	0	0	0
SUS304 (Metal)	*E* (Pa)	2.01 × 10^11^	0	3.08 × 10^−4^	−13.53 × 10^−7^	0
	*ρ* (kg/m^3^)	8166	0	0	0	0
	*α* (K^−1^)	1.23 × 10^−5^	0	8.09 × 10^−4^	0	0
	*K* (W/mK)	15.379	0	0	0	0
	*ν*	0.3177	0	0	0	0

**Table 2 materials-12-04084-t002:** Details of the four cases.

Material Parameters	Case 1	Case 2	Case 3	Case 4
Length *a* (m)	2	2	2	2
Width *b* (m)	2	2	2	2
thickness *h* (m)	0.1	0.1	0.05	0.1
Temperature *T_c_* (K)	500	500	500	800
Power law index *n*	1	2	1	1
The external pressure *Q* (N/m^2^)	2 × 10^4^	2 × 10^4^	2 × 10^4^	2 × 10^4^

**Table 3 materials-12-04084-t003:** Fundamental frequency parameter $\bar{\beta }=\omega \pi^{2}(a^{2}/h)\sqrt{\rho_{m}/E_{m}}$ of FGM plates for various boundary conditions and different power law indexes (*b*/*a* = 1, *h*/*a* = 0.01).

Boundary Conditions	n	m	Baferani [39]	Present	Error (%)
S-S-S-S	0	1	115.8695(1,1)	115.9250	0.048
		2	289.7770(1,2)	289.7708	0.002
	0.5	1	98.0136(1,1)	98.1594	0.149
		2	245.3251(1,2)	245.3680	0.017
	1	1	88.3093(1,1)	88.4500	0.159
		2	221.0643(1,2)	221.0950	0.014
	2	1	80.3517(1,1)	80.4195	0.084
		2	200.8793(1,2)	201.0104	0.065
S-C-S-C	0	1	170.0196(1,1)	170.0270	0.004
		2	321.4069(1,2)	321.5180	0.035
	0.5	1	143.8179(1,1)	143.9700	0.106
		2	272.1090(1,2)	272.2440	0.050
	1	1	129.6496(1,1)	129.7290	0.061
		2	245.1310(1,2)	245.3130	0.074
	2	1	117.8104(1,1)	117.9460	0.115
		2	222.8111(1,2)	223.0280	0.097
S-S-S-C	0	1	138.7717(1,1)	138.8740	0.074
		2	303.3569(1,2)	303.4670	0.036
	0.5	1	117.4222(1,1)	117.5913	0.144
		2	256.7762(1,2)	256.9595	0.071
	1	1	105.7770(1,1)	105.9598	0.173
		2	231.3509(1,2)	231.5406	0.082
	2	1	96.2668(1,1)	96.3352	0.071
		2	210.3895(1,2)	210.5070	0.056
S-S-S-F	0	1	68.5125(1,1)	68.6417	0.189
		2	162.8384(2,1)	163.1933	0.218
	0.5	1	58.0318(1,1)	58.1223	0.156
		2	137.9954(2,1)	138.183	0.136
	1	1	52.2092(1,1)	52.3732	0.314
		2	124.2452(2,1)	124.5145	0.217
	2	1	47.5511(1,1)	47.6162	0.137
		2	112.9582(2,1)	113.2039	0.218

**Table 4 materials-12-04084-t004:** Computation Times (CT) of the proposed method, Navier’s method, and FEM (Intel(R) Core(TM) i5-8250U CPU @ 1.6 GHz 1.80 GHz 8.0 GB (RAM), Mathematica 11.0, COMSOL 5.3 a, computation times in seconds).

Cases	CT of the Proposed Method (s)		CT of Navier’s Method (s)	CT of FEM (s)
Case 1	*N* = 7	174.23	281.64	445
	*N* = 9	287.92		
Case 2	*N* = 7	187.45	283.23	510
	*N* = 9	316.52		
Case 3	*N* = 7	169.32	278.53	468
	*N* = 9	304.77		
Case 4	*N* = 7	177.23	287.55	498
	*N* = 9	314.06		

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
