# Peer review of "Novel Semi-Analytical Solutions for the Transient Behaviors of Functionally Graded Material Plates in the Thermal Environment"

_materials, 2019, doi:10.3390/ma12244084_

Round 1

Reviewer 1 Report

Summary:

The authors present a semi-analytical algorithm, using Laplace transform, Fourier series expansion and the differential quadrature method for analyzing the transient response of functionally graded materials accounting for displacement and temperature variations. The method proposed by the authors is shown to be accurate and achieved fast convergence.

The paper is written clearly except some misspellings, non-descriptive figure captions, unclear figures and incomplete data.

Comments:

1. "Algorithm" is misspelled on several places. Please correct.

2. "FGM" should be explicitly described in the abstract.

3. At the end of the introduction (lines 96-101), the authors refer to the sections as "chapters". Please use "sections".

4. Figure captions should be more descriptive and should contain explanation of all the sub-figures.

5. What are t1 and t2 in equations 3 and 4?

6. Figure 2 is impossible to read. Please make use of different styles and larger font. And please add how much the value of N need to be in order to obtain an accurate solution.

7. Why are the boundary conditions for S-S-C-C and S-S-S-F not presented in Figures 5 and 6?

8. How long was each step used in the convergence studies? A key information missing is comparison with the convergence studies of other algorithms? Please also comment what kind of computer system was the algorithm run on.

9. It seems that although the difference between this study and Baferani and Jomehzadeh are very small, the results from this study almost invariably give a higher fundamental frequency except in the case where n = 0 and m = 2 for the SSSS case. Please comment if this error would be significant in cases where a design is based on determining the resonant frequency of the FGM plate.

10. What is the relationship between peak deflection and temperature? Is it linear or non-linear as here only two points at T = 500 K and T = 800 K are given?

Reviewer 2 Report

Authors are encouraged to improve their work by addressing the following remarks:

To mention some real materials whose thermal and mechanical properties are described by Eqs. (1) and (8). To discuss the effect of the exponent "n" in Eq. (1) on the temperature profile described by Eq. (5). To plot the errors generated by the proposed semi-analytical approach. To improve the English writing throughout the manuscript.

Reviewer 3 Report

  Factual comments: - It should be pointed out the advantage of the semi-analytical method against the Finite Element Method. It is stated that the proposed method is efficient but I did not find the comparison with (for example) FEM.   - The description of the numerical model is somewhat confusing. The excitation is described as f(x,y,t) = QSin(30t). It is not clear to which side (or part) of the sample is the excitation applied. Maybe a picture could help. The symbol Q is not described near its usage. There should be a reference to Table 2.   - It would be nice to present the source of the material properties. It is surprising, that density and thermal conductivity does not depend on temperature.   - It could be explained why the used mathematical apparatus (numerical Laplace transform) is not too restricting. The original problem is nonlinear.   - I would suggest drawing boundary conditions into a picture.     Formal comments: The language could be improved, the paper contains typos. It would be nice to use the same symbols (fonts) in the text and in equations. Using symbols is not consistent. Some figures (5, 6, ... ) are hardly legible. The symbol L from the description (line 193) is not in the equation (27). The style of references is not consistent.

Round 2

Reviewer 1 Report

The authors have made the changes and provided reasonable explanations to my comments.

Reviewer 3 Report

After revision, I can recommend the paper for publishing.